# Severe urinary tract damage secondary to primary bladder neck obstruction in women

**Pedro F. S. Freitas**[1]◉, **Augusto Q. Coelho**[1]‡, **Homero Bruschini**[1]‡, **Eric S. Rovner**[2]‡, **Cristiano M. Gomes**[1]◉*

1 Department of Urology, University of Sao Paulo School of Medicine, Sao Paulo, SP, Brazil, 2 Department of Urology, Medical University of South Carolina, Charleston, SC, United States of America

◉ These authors contributed equally to this work.
‡ AQC, HB and ESR also contributed equally to this work.
* crismgomes@uol.com.br

**Data Availability Statement:** All relevant data are within the manuscript and its Supporting Information files.

## Abstract

### Objective

To present the clinical and radiological characteristics of women with severe structural deterioration of the bladder and upper urinary tract secondary to Primary Bladder Neck Obstruction (PBNO), and their outcomes after bladder neck incision (BNI).

### Methods

Retrospective evaluation of adult women who underwent BNI for PBNO at one institution. Patients were assessed for symptoms, renal function, structural abnormalities of the urinary tract and video-urodynamics. PBNO diagnosis was confirmed with video-urodynamics in all patients. BNI was performed at the 4–5 and/or 7–8 o'clock positions. Postoperative symptoms, PVR, uroflowmetry and renal function were evaluated and compared to baseline.

### Results

Median patient age was 56.5 years (range 40–80). All presented with urinary retention–four were on clean intermittent Catheterization (CIC) and two with a Foley catheter. All patients had bladder wall thickening and diverticula. Four women had elevated creatinine levels, bilateral hydronephrosis was present in five (83.3%). After BNI, all patients resumed spontaneous voiding without the need for CIC. Median Qmax significantly improved from 2.0 [1.0–4.0] mL/s to 15 [10–22.7] mL/s (p = 0.031). Median PVR decreased from 150 to 46 [22–76] mL (p = 0.031). There were no postoperative complications. Creatinine levels returned to normal in 3/4 (75%) patients.

### Conclusion

PBNO in women may result in severe damage to the bladder and upper urinary tract. Despite severe structural abnormalities of the bladder, BNI was effective in reducing symptoms and improving structural and functional abnormalities of the lower and upper urinary tract.

**Funding:** The authors received no specific funding for this work.

**Competing interests:** The authors have declared that no competing interests exist.

## Introduction

Primary Bladder Neck Obstruction (PBNO) is an uncommon cause of bladder outlet obstruction (BOO) and lower urinary tract symptoms (LUTS) in women. The diagnosis requires a high index of suspicion by virtue of its uncharacteristic clinical presentation [1–4]. Women with PBNO may present with a combination of storage, voiding, and postmicturition LUTS, while some develop recurrent urinary tract infections [5–8]. A few series also report acute urinary retention in up to 16.6% of the patients [7, 9–11].

Severe bladder and upper urinary tract deterioration have been rarely reported among women with PBNO. Zhang et al. presented the largest series of PBNO in women [7]. Among 84 patients, the only structural abnormality reported was hydronephrosis due to vesicoureteral reflux in three patients. They did not report on structural abnormalities of the bladder nor renal insufficiency secondary to bladder dysfunction. Kumar et al. reported renal function impairment in six of 24 women in their series [12]. All patients had complete renal function recovery following catheter drainage, yet no data was provided on the structural characteristics of the bladder and upper urinary tract. In our literature review, we found no studies reporting on structural damage of the bladder and upper urinary tract due to PBNO in women.

Our recent experience with women presenting with bladder and upper urinary tract decompensation due to PBNO prompted us to review our experience with the treatment of this condition, including the clinical and radiological characteristics and the outcomes of bladder neck incision. We hypothesized that PBNO may include a spectrum of more severe presentation with damage to the bladder and upper urinary tract and sought to determine whether surgical treatment might benefit women with such features.

## Patients and methods

We reviewed our Electronic Medical Database for women $\geq$ 18 years who underwent Bladder Neck Incision (BNI) due to PBNO between September 2009 and July 2019. The present study protocol was approved by the Institutional Review Board of Hospital das Clinicas, University of Sao Paulo, under the protocol #13997. A Consent form was waived as this was a retrospective series based on an electronic medical database. PBNO diagnosis was confirmed with video-urodynamics in all patients based on recommended criteria of low flow and high detrusor pressures along with the narrow appearance of the bladder neck during voiding [8]. They also underwent cystoscopy in order to exclude bladder neck contracture or fibrosis. Exclusion criteria comprised congenital anomalies, pelvic organ prolapse, previous sling or pelvic surgery, and any evidence of neurologic or systemic disease with a potential impact on the lower urinary tract function.

Each case was evaluated in terms of clinical symptoms at presentation, use of medications with a possible effect on the lower urinary tract, renal function, urinary tract imaging, and video-urodynamic findings. Women with hydronephrosis or impaired renal function were initially managed with Foley catheter drainage. Laboratory, imaging studies, and video-urodynamics were performed after 3–4 weeks. Clean intermittent catheterization (CIC) was offered to women on chronic urinary retention after normalization of renal function.

All patients in this series were treated with BNI since they had severe complications of chronic obstruction including urinary retention, bladder diverticula, severe bladder wall thickening, vesicoureteral reflux (VUR), hydronephrosis and/or impaired renal function.

BNI was performed with an adult 24F resectoscope using a Collings knife to do one or two incisions at the surgeon's discretion, with the depth of the incisions limited to the circular muscle fibers, without reaching the perivesical fat. Incisions were made either at the 4–5 or 7–8 o'clock position or both, if bilateral. A 22F Foley catheter was placed at the end of the

procedure and typically removed at postoperative day 2 or 3. Patients were discharged once they were able to void spontaneously following catheter removal.

Clinical outcomes, PVR, free uroflowmetry, and complications were assessed for all patients on follow-up. LUTS after BNI were assessed by asking the patients whether they were globally satisfied with their voiding at each follow-up visit. Urinary tract imaging was repeated in those with significant preoperative abnormalities. Repeat urodynamics was reserved for patients whose LUTS persisted or recurred.

## Statistical methods

Quantitative data were expressed as medians and ranges while qualitative variables were expressed as absolute values, percentages, or proportions. The Wilcoxon matched-pairs signed rank test was used to compare continuous variables, and the Fisher's exact or chi-square test was used for categorical comparisons. All tests were 2-sided with p <0.05 considered statistically significant. Analysis was performed using commercially available statistical software (GraphPad Prism, version 8.03 for Windows, San Diego, California, USA).

## Results

A total of six women met the inclusion criteria. Median patient age was 56.5 years (range 40–80 years), and the median time between symptom onset and PBNO diagnosis was 7.5 years (range 1.5–34 years). All patients presented with urinary retention, including four on CIC, one with a Foley catheter and one with acute urinary retention in whom a Foley catheter was placed. Four women had elevated serum creatinine levels at presentation, including one with end-stage renal failure on hemodialysis. None of the patients had a history of continuous use of antimuscarinics, alpha-adrenergic agonists and/or sympathomimetic drugs such as norepinephrine reuptake inhibitors or tricyclic antidepressants.

Their baseline characteristics are shown in Table 1.

The video-urodynamic (VUDN) findings are described in Table 2. Pressure-flow studies confirmed bladder outlet obstruction with markedly elevated detrusor pressures in all but one patient. She had spontaneous voiding with low flow (Qmax = 7 mL/s) in free uroflowmetry and high post-void residuals, yet was unable to void or produce a detrusor contraction after three attempts of intubated flow. Nonetheless, a diagnosis of PBNO was established based on her severe LUTS combined with indirect evidence of bladder outlet obstruction at the bladder neck, including low Qmax, elevated PVR, increased detrusor wall thickness (5 mm) with bladder diverticula, and a closed bladder neck on VUDN. Based on the data presented in Table 2, the mean (SD) bladder outlet obstruction index (BOOI) [13] was 124.6 (66.6) for this cohort.

**Table 1. Clinical characteristics at initial presentation.**

| Patient | Age | Symptom Duration (years) | Bladder Drainage | UTI | Urinary Retention | Creatinine (mg/dL) |
|---|---|---|---|---|---|---|
| 1 | 49 | 10 | Foley (inability for CIC) | No | Yes | 1.60 |
| 2 | 80 | 1.5 | CIC | Yes | Yes | 0.95 |
| 3 | 41 | 8.0 | CIC | Yes | Yes | 1.50 |
| 4 | 40 | 7.0 | CIC | Yes | Yes | 0.82 |
| 5 | 70 | 3.0 | CIC | No | Yes | 6.60 |
| 6 | 64 | 34 | Foley | No | Yes* | 6.40 |

**CIC**: Clean Intermittent Catheterization; **UTI**: Urinary Tract Infection.

* Acute urinary retention at initial presentation.

**Table 2. Videourodynamic findings at initial presentation.**

| Patient | Capacity (mL) | DO* | Compliance (mL/cmH2O) | Qmax (mL/s) | PdetQmax (cmH2O) | PVR (mL) | Bladder outlet appearance |
|---|---|---|---|---|---|---|---|
| 1 | 200 | Yes | 66.66 | 1 | 105 | 150 | Narrow BN |
| 2 | 120 | No | 13.33 | 1 | 70 | 120 | Narrow BN |
| 3 | 532 | No | 29.55 | 2 | 140 | 500 | Narrow BN |
| 4 | 130 | Yes | 4.06 | 3 | 240 | 80 | Narrow BN |
| 5 | 250 | No | 41.66 | 7[+] | 0* | 150 | Closed BN* |
| 6 | 197 | Yes | 6.56 | 2 | 84 | 187 | Narrow BN |

[+] Derived from free uroflowmetry as this patient was unable to void on intubated flow.

* The patient was unable to void on 3 attempts of intubated flow. The diagnosis of LUTS was based on the previous evolution of symptoms.

**DO:** detrusor overactivity; **BN:** Bladder neck.

Table 3 summarizes the features of bladder and upper urinary tract damage at presentation. All cases showed bladder wall thickening and diverticula, while bilateral hydronephrosis was present in five (83.3%). Fig 1 illustrates severe structural damage to the bladder in different patients along with the characteristic narrow appearance of the bladder neck at micturition. Fig 2 demonstrates bilateral upper urinary tract dilation which resolved three weeks after bladder drainage with a Foley catheter.

Serum creatinine levels returned to the normal range in all but one patient. One woman (patient 5) presented with end-stage kidney disease in the absence of hypertension, diabetes, or any other identifiable risk factor. She was already on CIC and on hemodialysis when referred to our center for evaluation of her severe and bothersome LUTS. She initially refused surgery and opted for a trial of doxazosin, resulting in modest improvement. Her LUTS became more severe after a kidney transplant and resulting increase in urine output, and she underwent BNI.

BNI resulted in major symptom relief and a significant improvement in the pressure-flow parameters (Table 4). Median [IQR] Qmax significantly improved from 2.0 [1.0–4.0] mL/s to 15 [10–22.7] mL/s (p = 0.031), while median [IQR] PVR decreased from 150 [110–265] to 46 [22 – 76] mL (p = 0.031) after the procedure. There was no bleeding, urinary tract infection, or vesicovaginal fistula in the postoperative period. All patients successfully resumed spontaneous voiding once their catheters were removed, without the need for CIC on follow-up. No patient complained of stress urinary incontinence, nor did we observe it on standing cough test performed on follow-up visits.

**Table 3. Baseline structural abnormalities of the bladder and the upper urinary tract.**

| Patient | Bladder | | Upper urinary tract | | |
|---|---|---|---|---|---|
| | Wall Thickening | Diverticula | Hydronephrosis | VUR | Parenchymal Atrophy |
| 1 | Yes | Yes | Bilateral | No | No |
| 2 | Yes | Yes | No | No | No |
| 3 | Yes | Yes | Bilateral | No | No |
| 4 | Yes | Yes | Bilateral | High-grade, left-sided | No |
| 5 | Yes | Yes | Bilateral | No | Yes |
| 6 | Yes | Yes | Bilateral | No | No |

**VUR:** Vesicoureteral reflux.

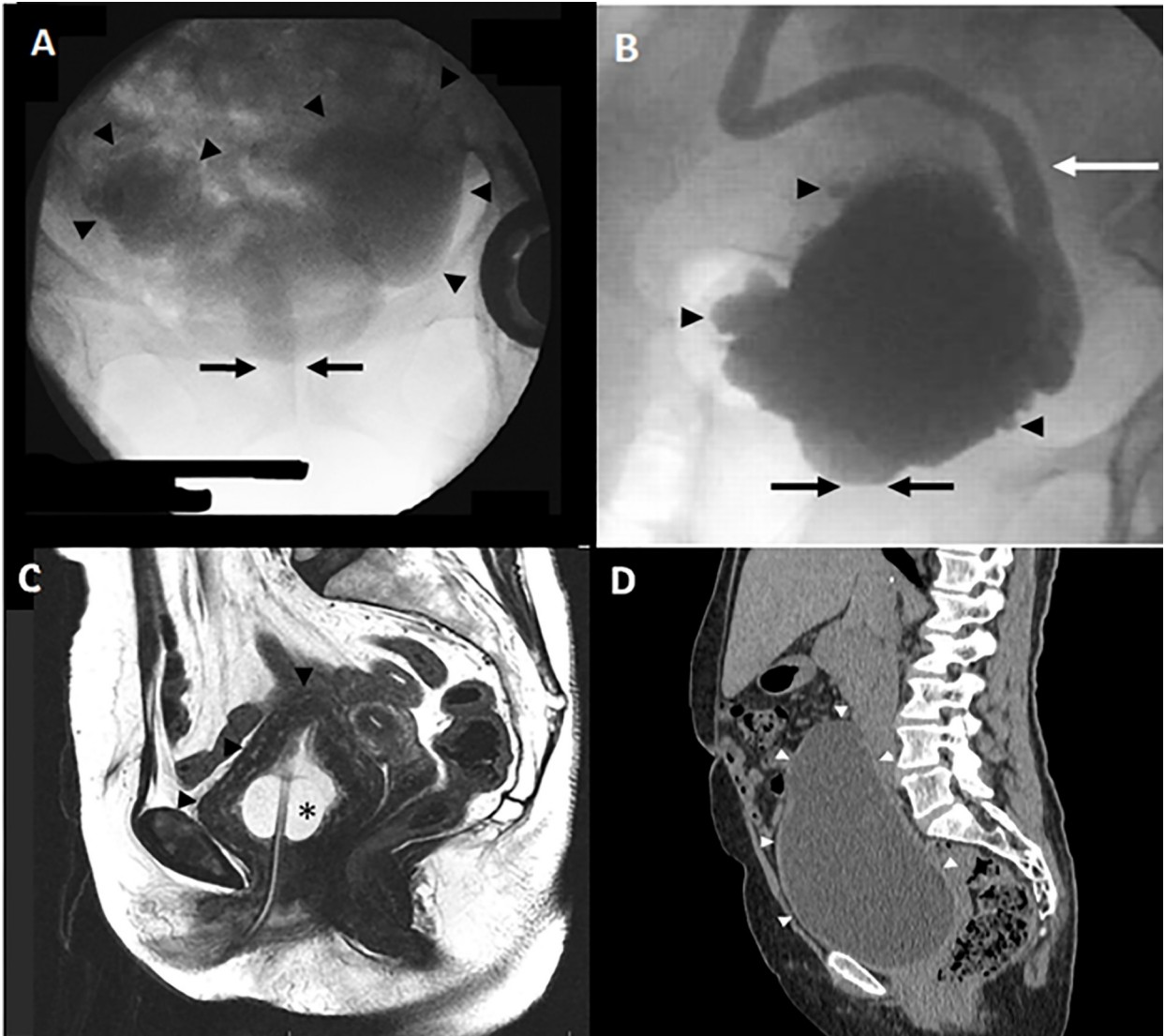

**Fig 1.** Baseline imaging studies illustrating bladder damage secondary to PBNO: **A)** Voiding cystourethrogram showing two large diverticula (arrowheads) and narrow (nonfunneling) bladder neck (arrows); **B)** Voiding cystourethrogram showing multiple diverticula (arrowheads), high-grade vesicoureteral reflux on the left side (white arrow), and narrow (nonfunneling) bladder neck (black arrows); **C)** Pelvic MRI on T2 weighted sequence showing severe bladder wall thickening (arrowheads) in the presence of a Foley catheter (asterisk). **D)** Non-contrast CT Scan (sagittal view) showing massive bladder distention (arrowheads) in a patient presenting with acute urinary retention.

One patient had recurrence of LUTS one year after her single incision bladder neck opening and eventually resumed CIC. A repeat VUDN confirmed BOO with a narrow bladder neck and another BNI was performed at two sites (5h and 7h), resulting in sustained symptom improvement at 14 months of follow-up. At a median follow-up of 16 [9.6–32] months, all the patients were voiding spontaneously, with no need for CIC, and very pleased with their lower urinary tract condition.

## Discussion

We present a series of cases of PBNO in women who developed severe damage to the bladder and the upper urinary tract. They all had features of long-term severe bladder outlet

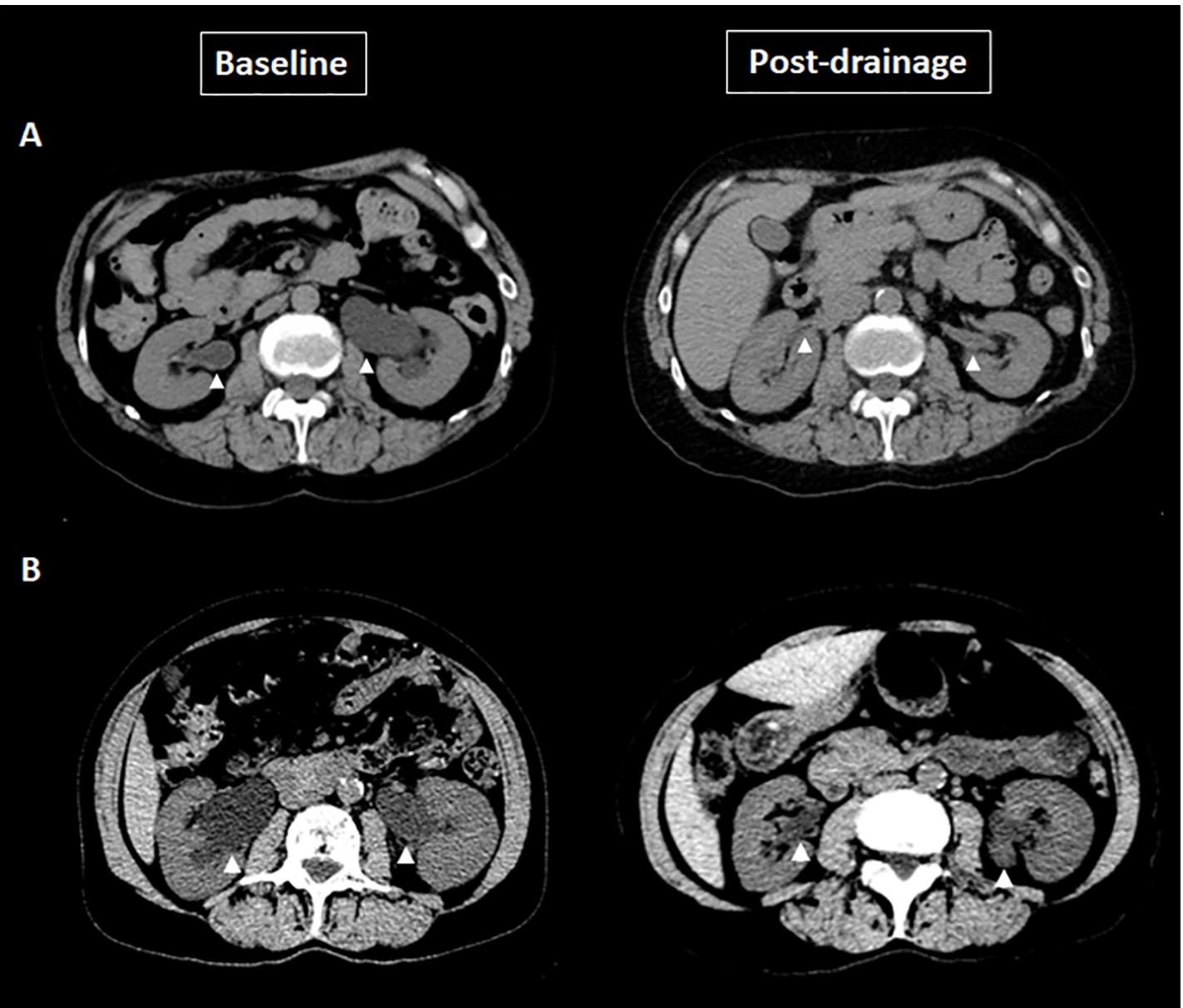

**Fig 2.** Non-contrast CT Scans illustrating severe upper tract deterioration secondary to PBNO: **A)** Bilateral hydronephrosis (arrowheads) with preserved renal parenchyma, resolved after three weeks with a Foley catheter. Her Creatinine levels decreased from 1.60mg/dL to 0.65mg/dL after drainage; **B)** Bilateral hydronephrosis (arrowheads) with preserved renal parenchyma, resolved after three weeks with a Foley catheter. Her Creatinine levels decreased from 6.60mg/dL to 0.67mg/L after drainage.

obstruction such as urinary retention, bladder diverticula, vesicoureteral reflux, bilateral hydronephrosis, and renal dysfunction. Despite the severity of these cases, BNI significantly improved LUTS in all patients without complications.

There are few studies on female PBNO, as this is an uncommon condition which has been described decades after PBNO in men [6]. Most series focus on the diagnostic features [2, 6] and treatment modalities [5, 7, 14, 15], with a paucity of data on the structural consequences of BOO to the bladder and upper urinary tract. Interestingly, we were not able to find data on urinary tract deterioration in men with PBNO as well [16, 17].

**Table 4. Impact of bladder neck incision on clinical and urodynamic variables.**

|  | Baseline (n = 6 pts.) | Post-operative (n = 6 pts.) | *p* value |
|---|---|---|---|
| **Qmax (mL/s)** | 2 [1 – 4] | 15 [10–20.7] | 0.031 |
| **PVR (mL)** | 150 [110–265.3] | 46 [22.5–76.2] | 0.031 |
| **Hydronephrosis** | 5 (83.3%) | 0 (0%) | 0.015 |
| **Impaired renal function*** | 3/5 (60%) | 0 (0%) | 0.167 |

**Qmax**: maximum flow rate; **PVR**: post-void residual volume.

* One patient removed from analysis because she was on hemodialysis and underwent kidney transplantation just before bladder neck incision; Qmax and PVR are expressed in medians and interquartile ranges.

Few series reported on structural damage to the urinary tract secondary to PBNO in women. Only two series reported on hydronephrosis: Zhang et al. [7] found it in three of their 84 patients, whereas Peng et al. (15) described it in one case. Kumar et al. [12] were the only to report data on renal function impairment which occurred in six of their 24 patients. Interestingly, all had complete functional recovery after bladder decompression, which was also the case in our series. Bladder diverticula have been inconsistently described in case reports [6, 18].

Based on these rare and anecdotal reports of structural damage to the lower and upper urinary tracts, one might be inclined to think that PBNO in women is a condition that almost only affect patients' quality of life, with a low potential for end-organ injury. Our findings, however, demonstrate that female PBNO may result in severe and irreversible damage to the bladder and the upper urinary tract.

The fact that all of our patients had severe structural abnormalities of the bladder and/or upper urinary tract may reflect a selected population in the worst spectrum of bladder neck obstruction. The patients included in this study represent the entire population of women who underwent BNI in our hospital in the past decade. With our procurement system, we were not able to retrieve patients based on the diagnosis of PBNO and it is probable that we missed less severe cases of PBNO in women that did not require BNI. It is also likely that mild cases might have been overlooked and not referred to our center, as this is a condition known to require a high level of suspicion for diagnosis.

Patients in our series would be classified as having overt bladder outlet obstruction by any of the different definitions of outlet obstruction in women [19–21]. For comparative purposes, we calculated the mean BOOI by using the average Qmax and Pdet values reported in the largest series of PBNO in women and found values ranging from 28 to 79 [2, 7, 10, 11, 22]. The mean BOOI of 124.6 observed in our cohort is thus consistent with a much higher degree of BOO. Such a severe degree of obstruction may be one of the reasons why our patients had such unusual and severe damage to the urinary tract.

Another explanation for such severe cases could be that they were evaluated and treated belatedly due to its uncharacteristic clinical presentation and rarity of this condition. The access of patients to specialized health care and their referral to our tertiary center may have been delayed by the obstacles of an overloaded public health system. In fact, the median seven-year time between onset of LUTS and the diagnosis of PBNO in our series is consistent with these hypotheses and may have contributed to the severity of structural abnormalities in our patients. Unfortunately, we were not able to find data on duration of LUTS in other series to compare with our findings.

Alpha-blockers have been successfully used to alleviate LUTS and improve urinary flow rates in women with PBNO but many patients require bladder neck incision, which is

considered the gold standard therapy, with success rates of 83% to 100% [5, 7, 10, 11, 22]. So far, there appears to be no consensus on the best technique for BNI in women, yet the larger and most recent series [7, 10, 22] have used two incisions at the 4–5 and/or 7–8 o'clock positions. We favor a single initial incision at either position, while the second is performed if the bladder neck does not appear to be enough open on intraoperative judgment. This less aggressive approach is favored by other experts [8, 11]. Our data show that BNI may be effective even when there is severe damage to the bladder. It consistently improved symptoms, uroflowmetry, and PVR of all our patients, with only one requiring a successful redo procedure.

Our study has several limitations. First, its retrospective design may lead to selection bias in terms of disease severity and treatment outcome. The low number of patients is also an important limitation, which is in part explained by the rarity of this condition [6]. Nevertheless, we present evidence that the natural history of PBNO in women may include a spectrum of a severe condition leading to the irreversible deterioration of the bladder and upper urinary tract, that the existing literature had not shown properly. Our findings should highlight the need for a high index of suspicion for the diagnosis of PBNO in women and also to exert extreme caution when adopting a symptom-based management. We hope that future studies on female PBNO will systematically report on the impact of this condition in the bladder and the upper urinary tract, which will clarify both the natural history and severity spectrum of PBNO in the female population.

## Conclusions

PBNO in women carries a potential for damage to the bladder and upper urinary tract, which may be secondary to either a higher degree of BOO or a delay in diagnosis and treatment. BNI was effective even in the presence of severe structural abnormalities of the bladder in reducing symptoms and improving structural abnormalities of the lower and upper urinary tract.

## Supporting information

**S1 Data.**
(XLSX)

## Acknowledgments

The authors are thankful to Dr. Daniel P. Magalhaes and Dr. Jose V. R. Garcia for their inestimable assistance in the preparation of the manuscript.

## Author Contributions

**Conceptualization:** Pedro F. S. Freitas, Augusto Q. Coelho, Cristiano M. Gomes.

**Data curation:** Pedro F. S. Freitas, Augusto Q. Coelho.

**Formal analysis:** Pedro F. S. Freitas, Cristiano M. Gomes.

**Investigation:** Pedro F. S. Freitas, Augusto Q. Coelho, Cristiano M. Gomes.

**Methodology:** Pedro F. S. Freitas, Homero Bruschini, Eric S. Rovner, Cristiano M. Gomes.

**Project administration:** Pedro F. S. Freitas.

**Supervision:** Homero Bruschini, Eric S. Rovner, Cristiano M. Gomes.

**Writing – original draft:** Pedro F. S. Freitas, Cristiano M. Gomes.

**Writing – review & editing:** Pedro F. S. Freitas, Homero Bruschini, Eric S. Rovner, Cristiano M. Gomes.

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
