## [Decision Letter · Decision Letter 0]

8 Jan 2021

PONE-D-20-39174

Severe urinary tract damage secondary to primary bladder neck obstruction in women

PLOS ONE

Dear Cristiano

Thank you for submitting your manuscript to PLOS ONE. After careful consideration, we feel that it has merit but does not fully meet PLOS ONE’s publication criteria as it currently stands. Therefore, we invite you to submit a revised version of the manuscript that addresses the points raised during the review process.

We look forward to receiving your revised manuscript.

Kind regards,

Israel Franco, M.D.

Academic Editor

PLOS ONE

Journal Requirements:

2. In the ethics statement in the manuscript and in the online submission form, please provide additional information about the patient records/samples used in your retrospective study, including: a) whether all data were fully anonymized before you accessed them; b) the date range (month and year) during which patients' medical records/samples were accessed; and c) the source of the medical records/samples analyzed in this work (e.g. hospital, institution or medical center name).

Reviewers' comments:

Reviewer's Responses to Questions

**Comments to the Author**

1. Is the manuscript technically sound, and do the data support the conclusions?

Reviewer #1: Yes

Reviewer #2: Yes

Reviewer #3: Yes

2. Has the statistical analysis been performed appropriately and rigorously? 

Reviewer #1: Yes

Reviewer #2: N/A

Reviewer #3: N/A

3. Have the authors made all data underlying the findings in their manuscript fully available?

Reviewer #1: Yes

Reviewer #2: Yes

Reviewer #3: No

4. Is the manuscript presented in an intelligible fashion and written in standard English?

Reviewer #1: Yes

Reviewer #2: Yes

Reviewer #3: Yes

5. Review Comments to the Author

Reviewer #1: This is a retrospective review of case series including a small number of women (6) presenting primary bladder neck obstruction and treated by bladder neck incision at one institution.

ABSTRACT: Abstract is well written and points out the most important results.

INTRODUCTION: Introduction is fine. The authors report important findings previously described on the subject and point out the gap in the literature as well as their objective with their study

METHODS: I believe that due to the retrospective aspect of the study the authors may have found in their database review only the most severe cases, that needed BNI, which may be a bias to the study.

What were the reasons to chose one or two bladder neck incision?

RESULTS: What do the authors attribute to be the reason for such a long period between the onset of symptoms and the diagnosis? It took more than 3 decades in one patient.

Do the authors did EMG analysis during uroflowmetry or UDS? That could have shortened the time between the onset of symptoms and the diagnosis.

Tables and figures are fine

DISCUSSION: The results on bladder outlet obstruction index (BOOI) should be described in the RESULTS section. Otherwise, discussion is fine.

After reading the paper I would suggest the authors to review the title, since they included only severe bladder neck obstruction, and that is what the paper is about. It would be more accurate if the title was something like this: Urinary tract damage secondary to SEVERE primary bladder neck obstruction in women

Reviewer #2: The authors present a well written and compelling argument for pnbd.

As some one who sees many young patients and adolescents with this problem and believes that it is a problems that persists into adulthood it would have been beneficial if there was some data regarding these women and their symptoms as children or adolescents to tie this condition to a long standing problem to further be able to increase awareness of this insidious problem.

if there are free flow stats on all patients it would be very useful to present this data as well so that the reader has a baseline from which to work prior to moving to invasive urodynamics. similarly what are the free flow numbers post incision.

We know from the pediatric literature that bn dysfunction is commonly associated with frontal lobe dysfunction and autonomic dysregulation. can you provide us with information regarding psychiatric diagnosis or disorders that the patients had and also if they were on any medications that would result in norepinephrine levels being elevated such as NERI, SSRI/NERI, sympathomimetics, Tricyclic antidepressants since these medications can exaggerate the response of the bn and create significant voiding issues.

Reviewer #3: This is a report of women with severe bladder neck dysfunction who underwent bladder neck incision. The authors found this procedure to be effective for patients with severe lower urinary tract dysfunction secondary to PBNO. It is clinically useful to know that although PBNO can lead to severe bladder damage, BNI is an effective treatment.

It is not clear in the introduction what is the authors' scientific hypothesis and research question. I understand that severe bladder changes have been poorly described for primary bladder neck dysfunction. However, the authors presented only two cystographic images with diverticulum and one MRI showing detrusor thickened. No image from the urodynamic study is presented. The CT image does not show thickened bladder. The hydronephrosis represented is mainly pelvis dilation.

As there are few cases presented by the authors it is important that more details of the cases are provided, such as the evolution of symptoms to urinary retention, history of urinary infection, urodynamics pictures and types of treatment before BNI.

The authors suggest hydronephrosis as an important finding of the study. However, in patients with urinary retention, bilateral renal dilation is the rule and not the exception. This is probably why this finding is not described extensively in the literature. Because it is an expected finding.

The authors comment that “Our findings, however, demonstrate that female PBNO may result in severe and irreversible damage to the bladder and the upper urinary tract.” However, although structural changes suggest severity, urodynamic is the most important finding for lower urinary tract evaluation. Video-urodynamic study is more important for the BNO diagnosis and for assessing vesicoureteral reflux than for diagnosing diverticulum, since its presence shouldn’t change the treatment.

How did the maximum detrusor pressure evolve after the procedure?

Was sacral stigma a diagnosis of exclusion? How has spina bifida occulta been ruled out?

In table 1 is creatinine at baseline or after CIC or after foley catheter placement? If it was at baseline, how many normalized with Foley?

How many used alpha blockers and for how long? Was standard urotherapy used and for how long? How about any type of electrical neuromodulation?

When there is no control group, as much information as possible before surgery should be given.

There is no indication where the discussion begins.

6. PLOS authors have the option to publish the peer review history of their article (what does this mean?). If published, this will include your full peer review and any attached files.

Reviewer #1: **Yes: **Jose Murillo B. Netto

Reviewer #2: **Yes: **israel franco

Reviewer #3: **Yes: **Ubirajara Barroso

---

## [Author Response · Author response to Decision Letter 0]

16 Feb 2021

The responses and clarifications to the points raised by the reviewers are within the 'Response to Reviewers' file attached to this submission.

---

## [Decision Letter · Decision Letter 1]

9 Mar 2021

Severe urinary tract damage secondary to primary bladder neck obstruction in women

PONE-D-20-39174R1

Dear Dr. Gomes,

We’re pleased to inform you that your manuscript has been judged scientifically suitable for publication and will be formally accepted for publication once it meets all outstanding technical requirements.

Kind regards,

Israel Franco, M.D.

Academic Editor

PLOS ONE

Additional Editor Comments (optional):

Reviewers' comments:

Reviewer's Responses to Questions

**Comments to the Author**

1. If the authors have adequately addressed your comments raised in a previous round of review and you feel that this manuscript is now acceptable for publication, you may indicate that here to bypass the “Comments to the Author” section, enter your conflict of interest statement in the “Confidential to Editor” section, and submit your "Accept" recommendation.

Reviewer #1: All comments have been addressed

Reviewer #2: All comments have been addressed

Reviewer #3: All comments have been addressed

2. Is the manuscript technically sound, and do the data support the conclusions?

Reviewer #1: Yes

Reviewer #2: Yes

Reviewer #3: Yes

3. Has the statistical analysis been performed appropriately and rigorously? 

Reviewer #1: Yes

Reviewer #2: Yes

Reviewer #3: Yes

4. Have the authors made all data underlying the findings in their manuscript fully available?

Reviewer #1: Yes

Reviewer #2: Yes

Reviewer #3: Yes

5. Is the manuscript presented in an intelligible fashion and written in standard English?

Reviewer #1: Yes

Reviewer #2: Yes

Reviewer #3: Yes

6. Review Comments to the Author

Reviewer #1: The authors have addressed all suggestions properly.

I still believe that the title should be revised.

Reviewer #2: no additional comments are necessary, all the questions were appropitely answered.

excellent study

Reviewer #3: The authors answered nicely all questions. The retrospective nature of the study is the main limitation, but do not invalidate the main findings.

7. PLOS authors have the option to publish the peer review history of their article (what does this mean?). If published, this will include your full peer review and any attached files.

Reviewer #1: No

Reviewer #2: **Yes: **Israel Franco, MD

Reviewer #3: **Yes: **Ubirajara Barroso Jr.

---

## [Editor Report · Acceptance letter]

11 Mar 2021

PONE-D-20-39174R1 

Severe urinary tract damage secondary to primary bladder neck obstruction in women 

Dear Dr. Gomes:

I'm pleased to inform you that your manuscript has been deemed suitable for publication in PLOS ONE. Congratulations! Your manuscript is now with our production department. 

Kind regards, 

on behalf of

Dr. Israel Franco 

Academic Editor

PLOS ONE